# Extracellular Vesicles from Lung Adenocarcinoma Cells Induce Activation of Different Cancer-Associated Fibroblast Subtypes

**DOI:** 10.3390/biomedicines12112523

**Published:** 2024-11-04

**Authors:** Jessica Angelina Trejo Vazquez, Rebecca Towle, Dylan Andrew Farnsworth, Masih Sarafan, William Wallace Lockwood, Cathie Garnis

**Affiliations:** 1Department of Integrative Oncology, British Columbia Cancer Research Center, Vancouver, BC V5Z1L3, Canada; jtrejo@bccrc.ca (J.A.T.V.); rtowle@bccrc.ca (R.T.);; 2Interdisciplinary Oncology Program, University of British Columbia, Vancouver, BC V5Z1L3, Canada; 3Division of Otolaryngology, Department of Surgery, University of British Columbia, Vancouver, BC V5Z1M9, Canada

**Keywords:** extracellular vesicles, lung adenocarcinoma, cancer-associated fibroblasts, tumor microenvironment, cellular communication

## Abstract

**Background:** Lung cancer, including the major subtype lung adenocarcinoma (LUAD), is the leading cause of cancer deaths worldwide, largely due to metastasis. Improving survival rates requires new treatment strategies and a deeper understanding of the mechanisms that drive tumor progression within the tumor microenvironment (TME). This study investigated the impact of extracellular vesicles (EVs) derived from LUAD cells on lung fibroblasts. **Methods:** EVs were isolated from LUAD cell lines via ultracentrifugation and characterized using nanoparticle tracking analysis and Western blotting. Lung fibroblasts were treated with PBS, TGFβ, or EVs, and their activation was assessed through protein (Western blotting) and RNA analyses (RNA seq and RT-qPCR). **Results:** The results confirmed the TGFβ induced activation and showed that LUAD EVs could also activate fibroblasts, increasing cancer-associated fibroblast (CAF) markers. While EV-induced CAF activation displayed unique features, like an increase in proliferation-related genes, the EV and TGFβ treatments also shared some differentially expressed genes. The EV groups induced a higher expression of ECM remodeling and EMT-associated genes, but some of those genes were also upregulated in the TGFβ group. Mesenchymal genes *POSTN* and *SPOCK1* were significantly upregulated in TGFβ- and EV-treated fibroblasts. Their secretion as proteins from the TGFβ- and EV-induced CAFs was not significant, confirmed through ELISA. **Conclusions:** These findings suggest that LUAD EVs play a role in CAF activation through both shared and distinct pathways compared to canonical TGFβ activation, potentially identifying novel gene expressions involved in CAF activation. Additionally, optimal protein secretion conditions of confirmed CAF-upregulated genes need to be established to determine their contribution to the TME.

## 1. Introduction

Lung cancer is the second most diagnosed cancer in both men and women and the most common cause of cancer deaths worldwide. Lung adenocarcinoma (LUAD) is the most common lung cancer subtype, accounting for ~40% of annually diagnosed cases [1]. The 5-year survival rate for people with LUAD is ~20% [2,3]. Most patients are diagnosed with advanced-stage disease due to the lack of available screening programs and the absence of clinical symptoms until advanced disease. Several diagnostic approaches can be used for LUAD detection, including X-ray, CT and PET imaging, and histological examination of tumor biopsies. Accurate staging of the cancer is required to determine the optimal treatment strategy. For Stage 0 or I, surgery, either a lobectomy or pneumonectomy is the first line of action, with a 5-year survival rate higher than 50%, and could be complemented with other therapies. However, adjuvant or neoadjuvant chemotherapy becomes more relevant during Stage II, with resection and additional radiotherapy [4]. In cases where surgery is not a viable option, radiotherapy can be used as an alternative treatment for patients. Adjuvant-targeted therapy aims at treating patients with specific mutations or translocations, including EGFR (osimertinib), ALK (alectinib), BRAF (dabrafenib and trametinib), and ROS1(entrectinib). In Stage III, the treatment options include surgery with neoadjuvant or adjuvant therapy, chemotherapy, and radiation therapy. Finally, during Stage IV, the main goal is to prolong survival and control symptoms; thus, treatments may include chemotherapy, radiotherapy-targeted therapy, and immunotherapy [1].

Even with recent treatment advances in recent years, prognosis is still poor. There is a clear need to develop new approaches for treating this disease [5]. A better understanding of the molecular basis of LUAD progression is needed to identify novel therapeutics that will increase survival rates.

Similar to many other cancer types, tumor progression in LUAD is influenced by the complex bidirectional communication between the tumor cells and host tissue within the tumor microenvironment (TME) [6]. The TME is a dynamic space composed of noncellular components, such as the extracellular matrix (ECM), and cellular components such as fibroblasts, epithelial cells, immune cells, inflammatory cells, and endothelial cells [7]. Bidirectional communication between cellular and noncellular components plays a role in tumor survival, settlement, growth, local invasion, and metastatic dissemination leading to LUAD progression [6,8,9].

One of the most abundant and functionally integral cell types in the TME is cancer-associated fibroblasts (CAFs) [10]. CAFs are a heterogeneous cell type and secrete a variety of molecules that promote ECM remodeling, induce the formation of blood and lymph vessels, drive drug resistance, and affect the immune response to enhance tumor growth and invasion [11]. Several CAF subtypes have been identified including Pan-myCAF, myofibroblast-like CAFs; pan-dCAF, desmoplastic CAFs; pan-iCAF and pan-iCAF-2, inflammatory-like CAFs; pan-nCAF, normal fibroblasts; and pan-pCAF, proliferating CAFs [12].

CAF heterogeneity can be partially attributed to the various activation pathways. Fibroblasts have been shown to be activated into CAFs by exposure to factors such as TGFβ1, PDGF, NF-κB, and HSF-1, which are secreted from nearby tumor cells or immune cells [13,14]. These factors can often be transported within cellular components of the TME through extracellular vesicles (EVs), lipid-bilayer-bound vesicles containing bioactive proteins, lipids, and nucleic acids capable of facilitating cell–cell communication [15].

Despite research outlining the ability of tumor-derived EVs to activate fibroblasts into CAFs [16,17], the characteristics of these CAF subtypes and their potential role in the LUAD TME have not been determined. Here, we aimed to gain a better understanding of the impact of LUAD-derived extracellular vesicles on neighboring fibroblasts and how the impacted fibroblasts may be contributing to tumorigenesis.

## 2. Materials and Methods

### 2.1. Cell Lines and Cell Culture Conditions

LUAD cell lines NCI-H1437 (ATCC^®^ CRL-5872™) and NCI-H2073 (ATCC^®^ CRL-5918™) were obtained from the American Type Culture Collection (ATCC, Manassas, VA, USA). The cells were grown in RPMI 1640 medium, which was supplemented with 10% FBS. Cancer cell lines were used within two months after being thawed to prevent genetic drift. Primary lung adult fibroblasts (H-6013) were purchased from Cell Biologics (Chicago, IL, USA); they were grown in a gelatin (Cell Biologics Cat # 6950) precoated tissue culture flask, and M2267 Complete Fibroblast Medium supplemented with 5% FBS was used. Primary fibroblasts were used within the 2nd and 10th passages. All the cells were cultured in an incubator at 37 °C with 95% humidity and 5% CO_2_.

### 2.2. Extracellular Vesicle Collection and Isolation

LUAD cell lines H1437 and H2073 were seeded in 15 cm plates. At ~30% confluency, they were washed with 1× PBS and cultured with RPMI 1640 media complemented with bovine EV-depleted 1% FBS. This process was conducted by the ultracentrifugation of FBS (110,000× *g*) for 16 h at 4 °C and then obtaining the supernatant. Before reaching 90% confluency, the media were collected to undergo a series of “slow spins” at 4 °C to remove cellular debris, which consisted of centrifuge cycles of 300× *g* for 10 min, 2000× *g* for 20 min, and 10,000× *g* for 30 min, recovering the supernatant in each step. Then, the supernatant underwent a series of “fast spins” to continue the isolation of EVs by precipitating them using an ultracentrifuge (Beckman Coulter, Fullerton, CA, USA) at 110,000× *g* at 4 °C for 90 min. After removing the supernatant, the pellet was washed with 1X PBS to re-pellet it using another 110,000× *g* for a 90 min cycle. Finally, the supernatant was carefully removed, and the EVs were resuspended in 0.02 μm-filtered PBS. The EVs were aliquoted to minimize freeze–thaw cycles and stored at −80 °C.

### 2.3. Nanoparticle Tracking Analysis (NTA) and Western Blot for EV Characterization

For NTA, the EVs were thawed, and the tube was thoroughly vortexed to later prepare a dilution of 1:2000 in 0.02 μm-filtered PBS for H1437. A dilution of 1:10,000 in H2073 was used to yield the optimal particle range (106–109 particles/mL), and the solutions were loaded into a Nanosight LM14C (Malvern Instruments, Amesbury, UK). Vesicle flow was analyzed for 60 s in triplicate per sample.

For WB analysis, the protein was collected from LUAD cells and their resuspended EVs. Samples were lysed by using a radioimmunoprecipitation assay (RIPA) buffer with 1:100 phosphatase inhibitor cocktail I and II (Sigma-Aldrich, St. Louis, MO, USA) and protease inhibitor cocktail Set III (Calbiochem, San Diego, CA, USA). Samples were incubated at 4 °C, separated from other contaminants, and a Pierce BCA Protein Assay Kit (Thermo Fisher Scientific, Norristown, PA, USA) was used to quantify the protein. After separating 10 μg of protein in a NuPage 4–12% Bis-Tris Gel (Thermo Fisher Scientific), the protein was transferred to a polyvinylidene difluoride membrane (Millipore, St. Louis, MO, USA). The membrane was blocked for 1 h at RT in a 1X TBS solution complemented with 5% BSA and 0.1% Tween-20. Then, they were incubated at 4 °C overnight with a dilution of blocking buffer and the corresponding primary antibody: 1:1000 anti-Alix (System Biosciences-53540, Palo Alto, CA, USA), anti-GRP 94 (sc-393402), anti-Hsp-70 (sc-24), anti-Flotillin-1 (sc-74566), and anti-CD81 (sc-166029) (Antibodies from Santa Cruz, CA, USA). After a series of 5 min washes, the membranes were incubated with the secondary horseradish peroxidase-conjugated antibody, either 1:2000 Anti-Rabbit (#7074, Cell Signaling, Danvers, MA USA) or Anti-Mouse (NXA931, GE Healthcare, Chicago, IL, USA). For detection, SuperSignal™ West Pico PLUS Chemiluminescent Substrate at 100% or SuperSignal™ West Femto Maximum Sensitivity Substrate (ThermoScientific) at 70% was used in a Chemidoc MP Imaging System (Bio-Rad, Hercules, CA, USA).

### 2.4. Coculture of Fibroblasts with EVs

H-6013 cells were seeded in 24-well plates with 1 mL of media and left to reach a confluency of ~15%. After 24 h, a quarter of the media was replaced with fresh media with 5% FBS, and the application of the treatments was initiated. This consisted of adding in duplicate every 24 h for 3 days per well: H1437 EVs (7.13 × 10^11^ particles), H2073 EVs (7.13 × 10^11^ particles), positive control TGFβ 1 (10 ng/mL) (R&D systems, reconstituted to a final concentration of 20 ug/mL), and negative control PBS. Protein (for WB) and RNA (for gene expression profiling) were harvested 48 h after the last treatment, which made for a total of 96 h of coculture.

### 2.5. Western Blot for CAF Activation

The previously mentioned protease inhibitors were added to RIPA buffer to prevent the degradation of the protein coming from the H-6013 samples 24 h after the third treatment of PBS or TGFβ or EVs in duplicate per group. The process of quantification, separation, and transfer was the same as that mentioned for the EV markers. After cutting the membrane into the sizes of the markers for this study, the fragments were blocked in 5% skim milk powder (instead of BSA) in 1X TBS and 0.1% Tween-20 for 30 min. Primary antibodies were diluted with 5% BSA blocking buffer, and the membrane was incubated overnight at 4 °C: 1:1000-diluted anti-PDGFRβ (3169s, Cell Signaling Technology, Danvers, MA, USA), anti-FAP (PA5-99313, Invitrogen, Carlsbad, CA, USA), anti-α-SMA (14968, Cell Signaling Technology), and 1:5000 anti-GAPDH rabbit (2118s, Cell Signaling Technology). The secondary antibody used was horseradish peroxidase-conjugated anti-rabbit antibody (7074S, Cell Signaling Technology) at RT for one hour in a dilution of 1:2000 for PDGFRβ, FAP, and α-SMA and 1:10,000 dilution for GAPDH in 2.5% skim milk powder in TBS-T. For detection, we used the same system described for the WB of EV characterization. We performed Western blot quantification using ImageJ version 1.53.

### 2.6. RNA Extraction and RNA Sequencing

RNA was extracted 48 h after the last addition of each treatment and control to H-6013 in duplicate; RNA was extracted using a miRNeasy Mini Kit (QIAGEN Inc., Germantown, MD, USA) following the protocol provided with the kit.

The samples obtained were sent to the Biomedical Research Centre at the University of British Columbia for quality control and sequencing. Approximately 100 to 200 ng was used as an input per sample, and the results of a 2100 Bioanalyzer Instrument (Agilent Technologies, Inc., Santa Clara, CA, USA) yielded RNA integrity numbers (RINs) of between 9.7 and 10. Samples fulfilling the quality requirements underwent the following process:

Qualifying samples were prepared following the standard protocol for the Illumina Stranded mRNA prep (Illumina, San Diego, CA, USA). Sequencing was performed on an Illumina NextSeq2000 with paired-end 59 bp × 59 bp reads. Sequencing data were demultiplexed using Illumina’s BCL Convert. Demultiplexed read sequences were then aligned to the Homo sapiens (hg38 no Alts, with decoys)/Mus Musculous (mm10) reference sequence using the DRAGEN RNA app on the Basespace Sequence Hub at https://support-docs.illumina.com/SW/DRAGEN_v41/Content/SW/DRAGEN/TPipelineIntro_fDG.htm (accessed on 13 October 2023) (SBME, 2023) [18].

### 2.7. Data Analysis of RNA Sequencing Data

RNA bulk sequencing analysis gave a result of around 40 million reads per sample. Assembly and differential expression were determined using DESeq2 at https://bioconductor.org/packages/release/bioc/html/DESeq2.html (accessed on 13 October 2023) in R. Log2 fold changes and adjusted *p*-values were used for the differential expression analysis comparing the treatment groups (EV or TGFβ) versus the control (PBS). PCA plots, batch corrections, and heatmaps were created in R using prcomp, ComBat_seq, and heatmap functions, respectively; for the volcano plots and the Venn diagrams, VolcaNoseR and Venny 2.1 were used respectively [19,20]. A *p*-adj < 0.05 and a log2 fold change > 1 were considered significant for bar plots and Venn diagrams and log2 fold change > 4 for the volcano plots. A *p*-adj < 0.01 and a log2 fold change > 3 were considered significant for the general heatmap. The heatmap of signature marker genes of specific CAF subtypes was created using the z-scores calculated from the normalized counts. Gene set enrichment analysis (GSEA) was performed by using a preranked list that considered the *p*-value and the log2 fold change in each treatment group, using a *p*-value of less than 0.05, an FDR of <0.25, and the Hallmarks gene set from MSigDB [21]. For GSEA, pathways with an FDR q-value < 0.25 and a nominal *p*-value < 0.05 were considered enriched. Pathways analysis from GSEA was visualized with R. ToppGene Suite (https://toppgene.cchmc.org/ (accessed on 13 October 2023)) was used to determine the functions of different gene sets.

### 2.8. Quantitative PCR

The RNA from the H-6013 cells treated with H1437 EVs, H2073 EVs, positive control TGFβ1, and negative control PBS (same samples sent for RNA sequencing) was put through TURBO DNA-free™ DNase Treatment to remove DNA contamination from the RNA sample. A total of 100 ng of RNA from each sample was used to conduct cDNA synthesis using the High-Capacity cDNA Reverse Transcription Kit (Applied Biosystems™, Foster City, CA, USA) following the manufacturer’s protocol. The cDNA sample from the different treatments underwent a preamplification reaction by following the Applied Biosystems™ TaqMan PreAmp Master Mix protocol. This was conducted using pooled primers for the target genes SPOCK1 (Hs00270274_m1) and POSTN (Hs01566750_m1) with GAPDH (Hs02758991_g1) as the control. Then, RT-qPCR was performed with an Applied Biosystems^®^ ViiA 7 Real-Time PCR System, and the expressions of the target genes from the activated fibroblasts treated with H1437 EVs, H2073 EVs, and positive control (TGFβ1_ were calculated with reference to the negative control (PBS) using the 2^−ΔΔCt^ method.

### 2.9. ELISA for EMT Candidates

H-6013 cells were cocultured with EVs for 96 h as previously described. Forty-eight hours after the third time EVs were added, the media were removed and replaced with serum-free media (without FBS). Forty-eight hours later, the conditioned serum-free media were collected and spun at 2000 rpm for 10 min at 4 °C; the supernatant was stored at −80 °C. Then, the cell culture supernatant was thawed, debris was removed, and the protein levels of SPOCK1 and POSTN were determined using enzyme-linked immunosorbent assays (ELISAs) (RayBiotech, Human Testican 1/SPOCK1 and Human Periostin ELISA Kits, Peachtree Corners, GA, USA) as per the manufacturer’s instructions. Analysis was performed per the manufacturer recommendations.

## 3. Results

### 3.1. EVs Isolated from LUAD Cells Are Mainly Small Extracellular Vesicles

To determine the influence of the EVs secreted by lung cancer cells on fibroblasts, we isolated the EVs from H2073 and H1437 lung adenocarcinoma cell lines through ultracentrifugation. Nanoparticle tracking analysis showed that the isolated particles had a diameter between 50 and 200 nm, indicating a predominance of small extracellular vesicles (Figure 1a,b). Western blotting was performed on the lysates of the cell lines and their corresponding EVs, revealing the expressions of the EV-associated proteins Alix, HSP70, Flotillin-1, and CD81 in the EV protein samples (Figure 1c). GRP94, also known as endoplasmin, is a protein primarily localized in the endoplasmic reticulum (ER) [17,22,23]. GRP94 is not usually present in small EVs; thus, it was used to evaluate EV purity and protein contamination. GRP94 could be detected in the cells but not in the isolated particles [24]. Taken together, we are confident that small EVs were the main component of the isolated pellets. These EVs were used for the subsequent experiments [25].

### 3.2. EVs from LUAD Cells Activate Fibroblasts into Cancer-Associated Fibroblasts (CAFs)

Markers, along with other criteria such as cell shape, location, and lack of lineage markers for other types of cells, can aid in identifying fibroblasts that are quiescent or activated [26]. CAF activation and recruitment can be mediated by the signals received from tumor cells in the form of paracrine factors including extracellular vesicles [27,28]. We hypothesized that the EVs from lung adenocarcinoma cells promote fibroblast activation via a distinct pathway.

To determine the impact of EV-induced CAF activation, primary adult lung fibroblasts (H-6013) were seeded and treated every 24 h with either PBS, TGFβ, or EVs (7.13 *×* 10^11^ particles) derived from H2073 and H1437 cells for three days.

Several positive-staining mesenchymal markers have been used to aid in identifying activated fibroblasts. Among the most common are α-smooth muscle actin (α-SMA), fibroblast activation protein (FAP), and platelet-derived growth factor receptor (PDGFR)-β [29,30]. Western blotting was performed to determine the changes in the expression of this protein panel (Figure 2a) [25]. TGFβ, a well-established activator of myofibroblast-like CAF subtype, was used as the positive control. There was a statistically significant increase in the α-SMA expression of the fibroblasts treated with TGFβ and H1437 EVs, with *p*-values of 0.021 and 0.001, respectively (Figure 2b). No significant difference in the fibroblasts treated with H2073 EVs was observed for the markers used when compared to the PBS group, our negative control (Figure 2b). It is known that the expression of CAF markers is widely heterogeneous and can significantly differ across CAF subtypes [28,29,30]. Traditional CAF markers, such as the ones we used, are not universally expressed by all CAFs: they can be highly expressed in some CAF subpopulations but not in others [31,32]. However, the upregulation of CAF activation markers was observed in all the groups in the RNA sequencing data (Table 1) [25]. To determine if TGFβ was also the CAF-activating factor for the EV groups, we assessed the protein lysates of the EVs (Figure 2c). A low expression of latent TGFβ was seen in the parental cells. However, it was not observed in their derived EVs. The activated form of TGFβ was not detected in either LUAD cells or their EVs, indicating that the EV transfer of TGFβ from LUAD cells is not the means of CAF activation.

### 3.3. EVs from LUAD Cells Induce CAFs with Shared and Distinct Phenotypes Compared to TGFβ

In order to gain insights into the mechanism of CAF activation by EVs, RNA was extracted from the fibroblasts after confirming their activation and was sent for RNA sequencing. Differential expression analysis was conducted using DeSeq2, which showed an upregulation of the CAF markers, including some of those markers that were used in the Western blotting, in fibroblasts cocultured with TGFβ and the EVs in relation to PBS control (Table 1). Overexpression was not seen for all markers in all samples or to the same extent, indicating differences in the function or activation of the CAFs [27,31]. We also observed that the fibroblasts treated with EVs had more differentially expressed genes (up- and downregulated) than the TGFβ group (Figure 3a). Considerable overlap of significantly differentially expressed genes was seen between the activation treatments, especially between the EV treatment groups for the upregulated genes (Figure 3b). Most of the significantly upregulated genes across the different groups were also very similar (Figure 4a–c) [25]. A heatmap was used to visualize the differences in the expressions of the most significant differentially expressed genes in the EV groups, with a log2 fold change higher than 3 and adjusted *p*-values lower than 0.01 (Figure 5). There were more genes with similar expression between the EV groups than with the PBS group, as expected, with more pronounced differences observed in the H1437 group compared to the H2073 group in relation to the PBS. group However, there were also differences between the two EV groups, with a subset of genes from the H2073 treatment resembling the PBS condition more than the other EV group (Figure 5, indicated in red box). This could be due to the differences between the cell lines and their cargo [33] (Figure 5).

### 3.4. LUAD EVs Induce Different CAF Subtypes and Upregulation of Proliferation, ECM Remodeling, and Mesenchymal Pathways

To determine the type of CAFs that were induced by the different treatments, we used the gene signatures of the CAF subtypes described by Galbo et al. [12]. Additionally, we used the Pearson correlation coefficient for this heatmap to calculate the distance between the columns for hierarchical clustering with the average linkage method (Figure 6) [25]. In the TGFβ-treated group, there was a higher expression of desmoplastic CAFs (dCAFs), especially the myofibroblast-like CAF (myCAFs) phenotype. For the H1437 EVs, these phenotypes were also increased (even more than in the TGFβ group); however, proliferative CAFs (pCAFs) were also observed with this EV treatment. For H2073-treated fibroblasts, we detected pCAFs as well and the presence of dCAFs and myCAFs to a lesser extent. We also observed the downregulation of inflammation-associated genes in our activated fibroblasts in all three treatment groups when compared to the PBS group.

To gain insights into the main function of the CAFs activated by the various treatments, we performed pathway analysis using GSEA. The commonly upregulated pathways in the three groups (compared to PBS) mainly included hypoxia, myogenesis, and glycolysis (Figure 7a–c). The biological pathways in common between the TGFβ and H1437 groups were mainly associated with TGFβ signaling and the downregulation of inflammation-related pathways (Figure 7a,b). The pathways in common between the EV groups consisted of upregulated gene sets associated with tumorigenic functions such as the cell cycle and EMT, as well as lung-cancer-related oncogenic pathways (Figure 7b,c).

### 3.5. Highly Upregulated Mesenchymal-Related Genes Are Secreted by CAFs

Based on the GSEA results, the three most deregulated pathways in the EV treatments were identical and not deregulated in the TGFβ activated group. These pathways involved tumorigenic functions affecting G2M checkpoint, E2F targets, and epithelial–mesenchymal transition (Figure 8a–f). The expression data for the genes associated with these pathways were visualized using volcano plots displaying the log2 fold change and the −log10 (*p*-adjusted) to show the extent of the differential expression of the genes that defined the pathway. The EMT pathway had the most upregulated genes in both EV-treated groups, with the highest log2 fold change values and the lowest *p*-adjusted values (Figure 8a,b). *SPOCK1* and *POSTN* were selected for the validation of the sequencing data. Interestingly, these genes were also upregulated in the TGFβ group, where the EMT pathway as a whole was not upregulated according to the GSEA analysis. Using qRT-PCR, we observed an increase of at least 15-fold and 13-fold in the activation treatments (TGFβ and EVs) compared to PBS for *POSTN* and *SPOCK1*, respectively (Figure 9a,b).

Confirming the sequencing data, PCR showed that *POSTN* expresion was significantly higher in the H1437 EV treatment than in the other treatments. *SPOCK1* expression was significantly higher in the H1437 EV and TGFβ treatments when compared to PBS. Finally, in the H2073 treatment, the expression of *SPOCK1* was significantly increased compared with that of PBS.

CAFs are known to release proteins that can impact the surrounding cells [34]. Recent research has revealed that CAFs in the TME are the primary source of EMT-related gene expression in the tumor parenchyma [35,36]. The two mesenchymal-related candidate genes, SPOCK1 and POSTN, are proteins that are secreted into the extracellular space [37,38]. In normal conditions, these proteins interact with cells and other ECM components to maintain the integrity and function of the extracellular matrix and aid in regulating cellular interactions [37,39]. However, their increased expression has been seen in malignant stroma in multiple cancer types and correlated with invasive characteristics [40,41]. Thus, their upregulation within activated CAF cells may also indicate increased secretion. To test this, SPOCK1 and POSTN proteins were evaluated through ELISA in the media collected from activated fibroblasts (Figure 10).

For POSTN, we observed more protein concentration in the three activation treatment media than in the PBS group, especially for fibroblasts treated with H1437-derived EVs. As mentioned before, POSTN expression was also higher in the induced CAFs (TGFβ and EVs) in relation to the negative control at the transcriptional level, especially in the EV groups (both in RNA-seq and qRT-PCR). This was despite small differences in the results of these two methods due to their different details, like technical differences, dynamic range, and normalization method. On the one hand, RNA-seq measures the entire transcriptome by sequencing all RNA molecules in a sample, while qPCR targets specific genes with designed primers [42,43]. Additionally, mRNA protein expression level correspondence has been notoriously poor in many studies [44]. As shown for periostin, we consistently found the gene expression in the H1437 EV group upregulated in all the stages, but, in this case, the increase was not significant in the media where its protein was secreted.

Similarly, for the expression of SPOCK1 in the media, there was more secretion from the activated groups (again, mainly in the H1437 group) than from PBS or even TGFβ. Interestingly, despite the high mRNA levels in all the activation groups, the protein secretion from the CAFs (mainly in the TGFβ group) was low. Several factors can affect the levels of a gene’s mRNA to match the protein levels observed after secretion, like the rates of transcription and translation, post-transcriptional regulation, post-translational modifications, and secretion and localization, where SPOCK1 may not have been entirely secreted at this point from the CAFs [44,45,46]. Thus, further study of this gene is required to elucidate its importance in the TME.

## 4. Discussion

The presence of cancer cells in the lungs can result in a chronic response to damage from stromal cells. These mesenchymal cells then become constitutively activated, resistant to apoptosis, and the origin of CAFs [45]. To achieve this, tumors can influence the surrounding cells through elaborate networks of direct cell–cell contact or by secreting EVs [46]. Some soluble mediators allow horizontal genetic/biomaterial transfer, like circulating tumor cells, circulating nucleic acids, and extracellular vesicles [47]. These intercellular communication mechanisms aid in understanding how the progression of cancer is facilitated by the tumor microenvironment [48].

Extracellular vesicles are a very heterogeneous group of particles categorized based on different characteristics, such as origin, size, and function. EVs can range from 20 nanometers to 10 microns [49]. If they are smaller than 200 nm, they are considered to be small EVs (sEVs), which include exosomes [50]. In our case, we isolated the EVs by ultracentrifugation, and, since it is increasingly clear that the isolates tend to be a heterogeneous population of EVs rather than one single type, we identified them based on their size range [33,51]. Therefore, we demonstrated that the most frequently occurring particle size of the samples obtained from the lung cancer cell lines H2073 and H1437 was within a range of 30–150 nm, specifically around 117 nm and 140 nm, respectively. This low particle size range corresponds to small vesicles, and, therefore, the particles isolated seemed to belong mainly to this category [52] (Figure 1). Different types of EVs impact their recipient cells differently due to the diverse cargo they contain, which can depend on the cell origin, biogenesis, size, and composition [33]. In fact, we saw that the EVs derived from the two LUAD cell lines showed similarities in their effects on the lung fibroblasts but also differences, which might have been due to the differences in their content and, thus, their influence on the recipient cells [53].

Our results show that all the treated fibroblast groups had a considerable amount of significantly upregulated genes in common, which seemed to be associated with the extracellular matrix’s structural constituent’s molecular function (Figure 3). However, the TGFβ group seemed to be more specific, having structurally associated functions and components of the extracellular matrix that are involved in the regulation of cell adhesion, migration, and tissue organization. These functions are relevant since the remodeled ECM generates physical barriers because it is denser and more rigid than normal tissue, and more resistant to drug penetration. This can also lead to hypoxia and an effect on blood vessels [54]. Corroborating the similarities observed between the groups, in Figure 4, several significant and upregulated genes are similar across the groups. Interestingly, the functions associated with the genes that are more specific to the EVs could impact cancer progression since the microtubules contribute to maintaining cell structure and shape, having an important role during cell division [55]. Both EV groups had a higher propensity for cell cycle progression and mitosis, as suggested by the GSEA (Figure 7). Therefore, the pathways of the TGFβ-treated cells were linked with extracellular matrix organization, while both EV groups were additionally involved in cell cycle mitosis and cell cycle checkpoints.

However, there were differences between the EV groups as well, since there was a higher expression of certain genes in the H1437EV group than in the H2073EV group and an even lower expression in the PBS-treated cells (Figure 5). The functions of these genes are associated with lung fibrosis. Other genes seemed to have a higher expression in the H2073 EV and PBS groups, with a very low expression in the H1437 EV group, as shown within the red box in Figure 5, along with other genes that resemble those in the PBS group. Interestingly, their functions are related to the inflammatory response and fatty acid transport and binding. These differences can be associated with differences in EV cargo, depending on the cell of origin and consisting of diverse molecules, such as nucleic acids. Different groups have studied the RNA content of tumor-derived EVs. Lawson et al. determined that EVs derived from LUAD cells showed higher expressions of a subset of miRNAs in comparison to the cells themselves and that the types of cargo from EVs derived from H1437 and H2073 have some differences [56].

CAFs are highly heterogeneous, with different subtypes exhibiting distinct functions. Interestingly, the ratio of specific CAF subpopulations has been linked to clinical outcomes [57]. Therefore, the identification and study of CAF subtypes may enable the development of prognostic markers and therapeutic targets, overcoming resistance and improving patient outcomes for different malignancies. For lung cancer, Galbo et al. individually analyzed the proportion of different CAF subtypes based on the study by Lambrechts et al., identifying four subtypes for this specific cancer [12,58]. The CAFs with the highest proportion were the dCAFs, representing 44.8% of the population, followed by myCAFs with 31.7%, iCAFs with 13.4%, and metabolic CAFs (mCAFs) with 10.1%. Importantly, Galbo et al. had similar results regarding the type of fibroblasts derived from the tumor to those found by Lambrechts et al., since they validated the gene expression associated with dCAFs, myCAFs and mCAFs in Lambrecht et al.’s data. Additionally, they believed that the iCAFs detected in Lambrecht et al.’s study could be from fibroblasts that were not tumorigenic due to the lack of expression of additional CAF-related genes [12]. This is interesting since our results also showed the phenotypes of myCAFs and dCAFs in our positive control and the EV-treated fibroblasts (each treatment with their corresponding population proportions), with the absence of or decrease in inflammation-related genes in the three treatments as well.

We identified the CAF subtypes associated with CAF activation by TGFβ or by LUAD-associated EVs. In agreement with the literature, in the TGFβ group, myofibroblast-like CAFs were observed [59] (Figure 6). myCAFs can be distinguished by the increase in genes involved mainly in extracellular matrix (ECM) organization and smooth muscle contraction [60]. In our study, while the three treatment groups showed a rise in α-SMA (ACTA2) and FAP, the H1437-treated group showed the highest expression levels. This suggested the enrichment of myCAFs, especially in the H1437 group, even when compared to the canonical TGFβ positive control. In addition, H2073-treated cells induced a myCAF phenotype but to a lesser extent when compared to the others. This corroborated the WB results, where α-SMA had significantly higher expression in H1437-treated fibroblasts than in H2073-treated fibroblasts, as shown in Figure 2. It has also been observed that myCAFs highly express genes related to angiogenesis [61]. Both EV-treated groups and the TGFβ-treated group showed positive enrichment of PDGFA and MEF2C, which is related to neo-angiogenesis, in addition to high levels of VEGF [12]. This suggested an increase in angiogenesis-related genes in the three treatment groups but especially in the TGFβ one. On the other hand, dCAs and pCAF populations have been associated with ECM remodeling and enhanced EMT in cancer [62]. We observed that the EV-treated fibroblasts had dCAF and myCAF signatures, demonstrating their similarity to the TGFβ-treated group. However, as mentioned before, the proportions of the phenotypes for these two subtypes would differ between H1437, H2073, and TGFβ. Therefore, this also shows the specifics of EV-induced CAF activation by different LUAD cell lines and TGFβ that we previously saw in the unique genes enriched in each treatment (Figure 3b). Thus, this could be the reason for the differences in some of their functions as well. Additionally, EV-induced CAFs were both enriched in proliferative CAFs (Figure 6). This suggested the progression to mitosis in the EV-treated group, consistent with our pathway analysis (Figure 7). In certain cancer types where dCAFs are associated with poor clinical outcomes, there was an increase in the gene sets related to the cell cycle, ECM, and collagen [63]. Furthermore, tumors with high amounts of dCAFs and pCAFs showed increased gene expression associated with the cell cycle, since pCAFs have increased proliferative and metabolic abilities, proving an enhanced replication potential in the TME [12,64]. Therefore, the EV cargo can play a role in the type of CAF that arises.

Knowing that CAFs signal back to the surrounding cells, we considered the top biologically relevant and statistically significant genes that are known secreted proteins, POSTN and SPOCK1 (Figure 8 and Figure 9). Periostin is a secreted extracellular matrix protein [38]. It has been shown that POSTN is highly expressed in malignant stroma, and CAFs can produce it to aid the development of the TME [65,66,67]. Yamato et al. demonstrated that activated fibroblasts in pulmonary fibrosis secreted periostin and promoted the progression of NSCLC, contributing to poor prognosis [68]. Similarly, Takatsu et al. found that the POSTN secreted by CAFs participated in EMT induction and drug resistance in NSCLC [69]. Our results confirmed that periostin expression was upregulated, with no significant secretion from the induced CAFs. In the case of TGF-β, studies demonstrated its consistent ability to induce periostin expression in a time-dependent and dose-dependent manner across different cell types [65,66]. The novel finding of this study is that it also happened after fibroblasts were activated with LUAD EVs, especially in the H1437 group. However, further timing and dose studies need to be conducted regarding its optimal secretion. Therefore, LUAD EV-mediated CAF activation might represent another mechanism for POSTN upregulation and presence in the TME.

SPOCK1 is a proteoglycan that fibroblasts can produce, and it can be eventually secreted into the extracellular space, where it regulates the equilibrium of the ECM [67]. However, it has been found that SPOCK1 is predominantly expressed in the stromal compartment of tumors, aiding in ECM remodeling, which is essential for tumor cell invasion [41,70]. SPOCK1 is involved in facilitating EMT in lung cancer, and its high expression has been correlated with malignant invasive characteristics, the formation of an immunosuppressive TME, and poor prognosis [71]. We found that its gene expression was upregulated in CAFs, but its protein concentration in the media was low. As mentioned, several factors could impact *SPOCK1* gene expression in the process of its mRNA turning into protein and being secreted. Importantly, the expression and secretion of SPOCK1 in CAFs have not been widely described in the literature nor has its mechanism. With these activation treatments and under these conditions, we observed signs of its secretion from the cells. Thus, further study of the optimal secretion conditions for this protein is required to determine its impact on the TME.

## 5. Conclusions

Our results suggest that EVs derived from LUAD cells induce CAF activation in fibroblasts. Additionally, these EV-treated CAFs have some unique features, such as an increase in proliferation-related genes, but they also seem to have similarities with the TGFβ treatment phenotype. However, these LUAD EVs do not contain TGFβ; thus, they do not transfer the different forms of the protein in their cargo. Several CAF subtypes have been identified, and a better understanding of their characteristics and functions could allow us to identify novel therapeutic targets, which may improve patient outcomes. Furthermore, EV activation treatments induce a high expression of ECM remodeling and EMT-associated genes in these CAFs. Highly expressed mesenchymal-related genes could be secreted from EV-induced CAFs in the TME and promote ECM remodeling along with stiffness while inducing the epithelial-to-mesenchymal transformation of the surrounding cancer cells. Both processes can facilitate the migration and invasion of lung cancer cells.

## Figures and Tables

**Figure 1 biomedicines-12-02523-f001:**
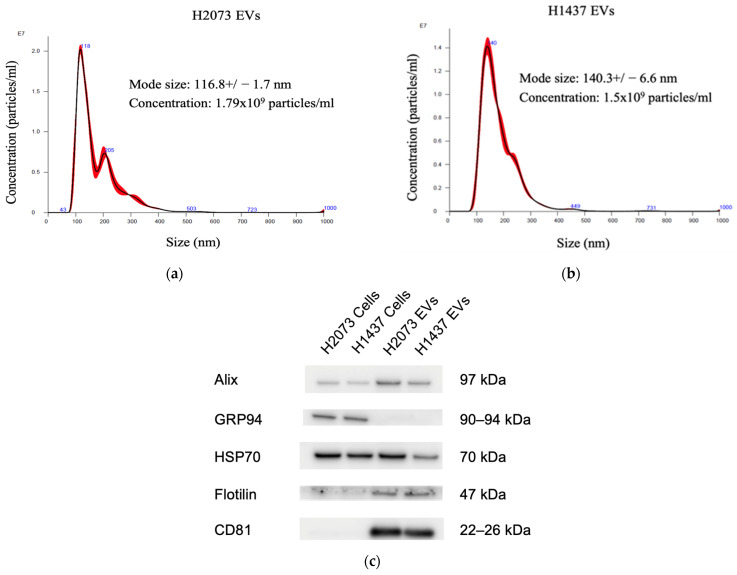
EV characterization. Nanoparticle tracking analysis for characterization and concentration of EVs with a NanoSight: (**a**) H2073 EVs, (**b**) H1437 EVs. (**c**) Western blot of the protein from LUAD cells and their EVs.

**Figure 2 biomedicines-12-02523-f002:**
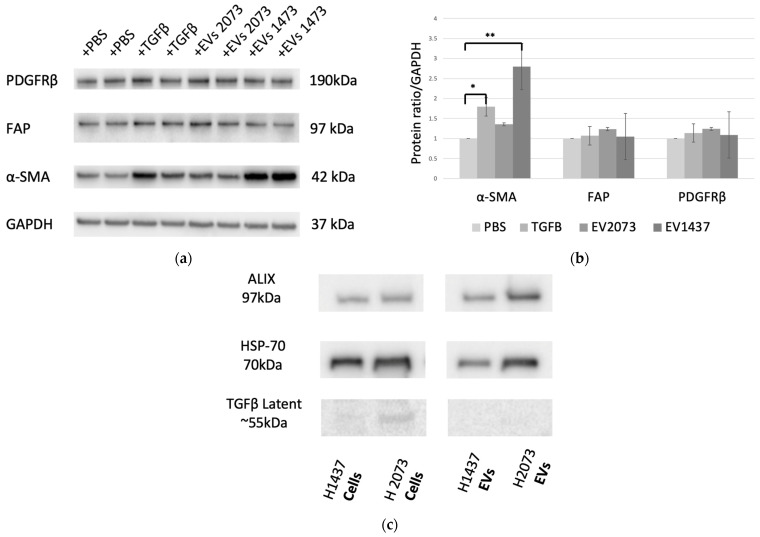
Characterization of CAF activation. (**a**) CAF activation by TGFβ and H1437 EVs (7.13 × 10^11^ particles added for both LUAD cell lines). (**b**) Western blotting quantification showed upregulation of markers in the TGFβ (α-SMA, *p*-value = 0.021) and H1437EV (α-SMA, *p*-value = 0.001) groups compared to the negative control. * *p* ≤ 0.05 and ** *p* ≤ 0.01. One-way ANOVA with the Holm–Sidak method. (**c**) Western blot measuring TGFβ1 content in LUAD cells and EV lysates.

**Figure 3 biomedicines-12-02523-f003:**
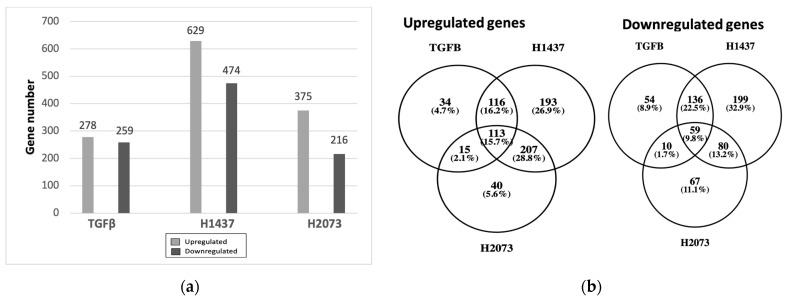
Differentially expressed genes in fibroblasts treated with TGFβ or EVs from H1437 and H2073 cells. (**a**) Bar graphs show the amount of upregulated and downregulated genes in each group. (**b**) Venn diagrams show the numbers and percentages of upregulated and downregulated genes in common between each group.

**Figure 4 biomedicines-12-02523-f004:**
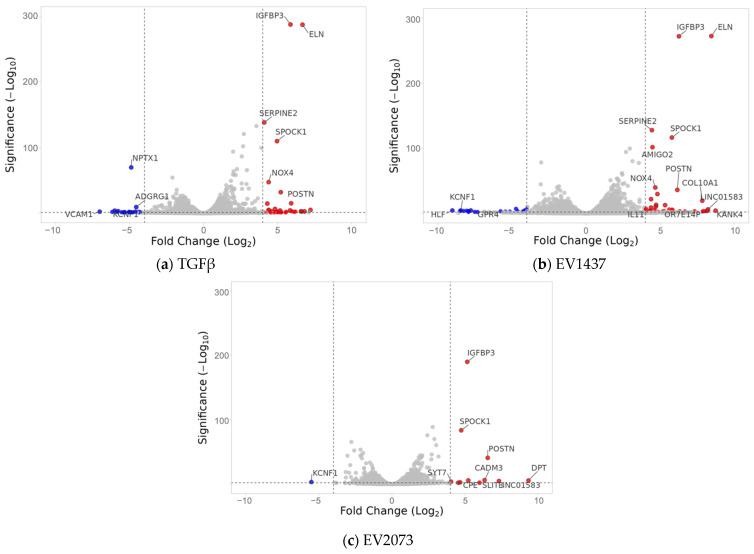
Volcano plots display the differentially expressed genes in the three treatment types compared to the PBS control. The top genes showing the highest differential expression are indicated. The red dots represent the significantly upregulated genes, and the blue ones are the significantly downregulated genes. The grey dots represent the genes considered unchanged. The dotted lines are the fold change threshold, which was four.

**Figure 5 biomedicines-12-02523-f005:**
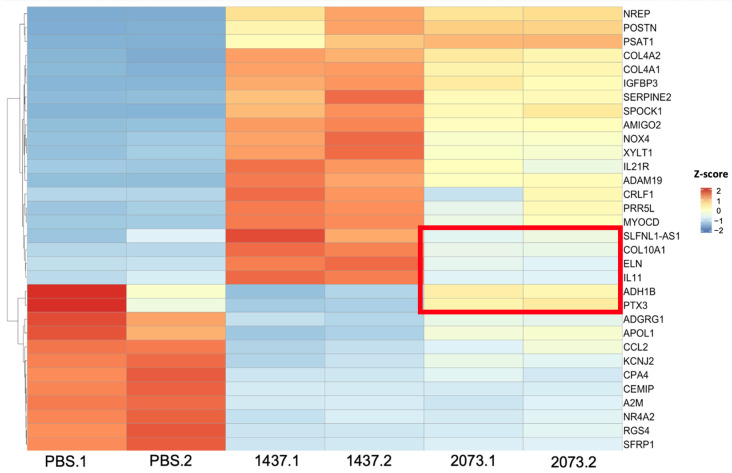
Most significant differentially expressed genes in EVs. Expression: high (red), middle (white), low (blue). Distinct genes in the H2073 EV group compared to the H1437 EV treatment are marked with a red rectangle. Z-scores were calculated from normalized counts in differential expression analysis of fibroblasts cocultured with EVs from H1437 and H2073 cells compared to fibroblasts cocultured with PBS.

**Figure 6 biomedicines-12-02523-f006:**
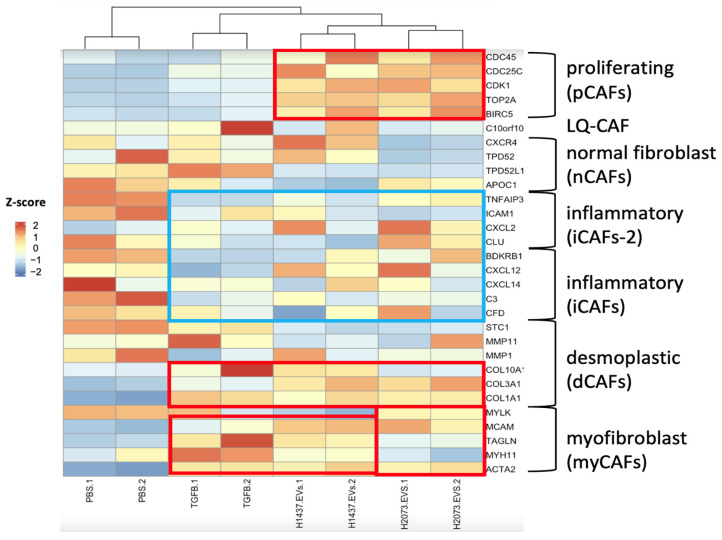
Specific marker genes related to pan-CAF subsets (right). Expression: high (red), middle (white), and low (blue). Higher expressions of marker genes linked to pan-pCAF, pan-dCAF, and pan-myCAF when compared to the PBS group are marked with red rectangles. Lower expressions of marker genes linked to pan-iCAF and pan-iCAF-2 when compared to the PBS group are marked with a blue rectangle. Z-scores were calculated from normalized counts in differential expression analysis of fibroblasts cocultured with TGFβ, EVs from H1437 and H2073 cells, compared to fibroblasts cocultured with PBS.

**Figure 7 biomedicines-12-02523-f007:**
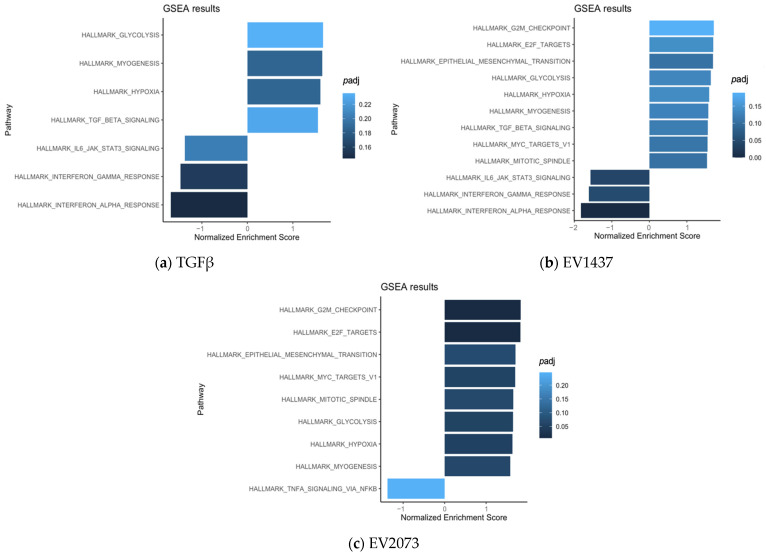
Pathways enriched in activated fibroblasts by TGFβ (**a**) and EVs (**b**,**c**).

**Figure 8 biomedicines-12-02523-f008:**
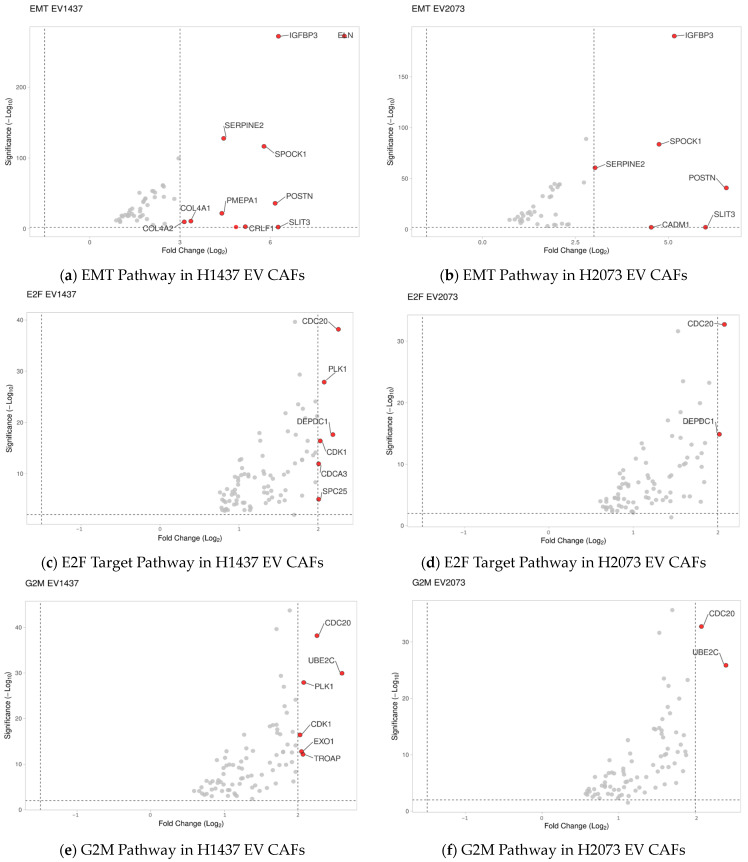
Volcano plots display the most significant changes in (**a**,**b**) EMT-related genes, (**c**,**d**) E2F-target-related genes, and (**e**,**f**) G2M-related genes. The red dots represent the significantly upregulated genes, and the grey dots represent the genes considered unchanged.

**Figure 9 biomedicines-12-02523-f009:**
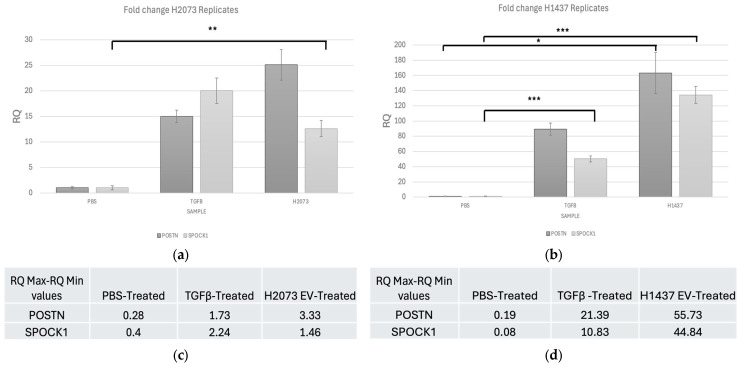
qRT-PCR shows fold change for *POSTN* and *SPOCK1* in activation treatments compared to PBS for (**a**) H2073 EV-treated fibroblasts. Upregulation of *SPOCK1* in H2073 (*p*-value = 0.002) group compared to the negative control. (**b**) H1437 EV-treated fibroblasts. Upregulation of *POSTN* in H1437 (*p*-value = 0.017) and of *SPOCK1* in H1437 (*p*-value < 0.001) and TGFB (*p*-value < 0.001) groups compared to the negative control. * *p* ≤ 0.05, ** *p* ≤ 0.01, and *** *p* ≤ 0.001. One-way ANOVA with the Holm–Sidak method using delta CT values. Error bars represent RQ min–RQ max values (**c**) H2073 EV-treated fibroblasts error bar values. (**d**) H1437 EV-treated fibroblasts error bar values.

**Figure 10 biomedicines-12-02523-f010:**
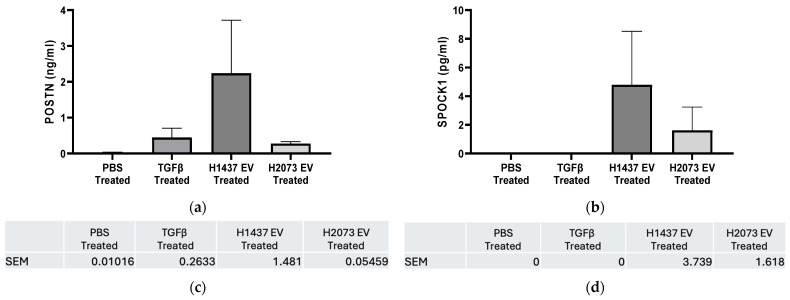
ELISA results for H2073 EV treatment and H1437 EV treatment. POSTN (**a**) and SPOCK1 (**b**) protein levels were measured in the cell culture supernatant of H-6013 fibroblast cells after 96 h of treatment with PBS, TGFβ, H1437 EVs, or H2073 EVs. Protein quantified using ELISA. Error bars represent SEM. No statistically significant difference was seen for POSTN and SPOCK1 in the groups compared to the negative control according to one-way ANOVA. (**c**) POSTN SEM error bar values. (**d**) SPOCK1 SEM error bar values.

**Table 1 biomedicines-12-02523-t001:** Differentially expressed CAF-associated genes and their fold changes that were used in identifying CAF phenotype in each treatment group.

Gene	+TGFβ	+H1437	+H2073
*COL1A1*	1.98	2.27	2.41
*COL1A2*	1.06	2.61	2.58
*COL3A1*	0.44	1.36	1.26
*COL4A1*	2.54	3.52	2.30
*COL4A2*	2.24	2.20	1.63
*COL5A1*	1.76	2.61	3.04
*COL6A1*	0.06	1.30	1.15
*COL7A1*	1.48	1.92	1.40
*COL8A1*	0.97	1.04	0.26
*COL10A1*	5.92	7.85	3.44
*COL11A1*	1.72	1.51	1.75
*COL13A1*	−0.45	−0.88	−1.06
*COL16A1*	0.81	1.30	1.22
*TAGLN*	1.08	1.26	0.39
*ELN*	6.69	8.46	2.73
*BGN*	2.67	2.95	2.79
*DCN*	−0.20	0.31	0.36
*LUM*	1.02	1.90	1.97
*MMP1*	−0.77	−0.02	−0.48
*MMP2*	1.42	1.79	1.37
*MMP3*	−1.37	−0.32	−0.83
*MMP10*	2.17	−0.57	2.08
*MMP11*	0.64	−0.47	0.58
*MMP14*	0.42	0.56	0.13
*MMP19*	0.80	0.50	0.80
*SERPINE1*	1.06	1.38	0.40
*CTHRC1*	2.06	2.17	1.89
*THBS2*	0.72	1.01	0.03
*SULF1*	0.50	0.56	−0.11
*COMP*	3.61	2.20	3.14
*INHBA*	1.54	1.78	0.11
*TGFBI*	1.56	2.43	2.08
*PDGFA*	1.33	1.31	0.06
*PDGFC*	1.14	1.01	0.73
*VEGFA*	0.41	0.27	0.50
*ANGPT2*	0.07	−1.28	−1.47
*ACTA2*	1.52	1.85	1.62
*FAP*	2.03	2.08	1.84
*POSTN*	5.21	6.16	6.56
*PDGFRβ*	0.55	0.17	0.19
*MYL6*	0.11	0.35	−0.27
*MYH9*	0.43	0.19	0.21
*MYH11*	0.94	0.72	−0.42
*TPM1*	2.01	2.46	1.07
*SORBS2*	5.37	6.14	3.38
*RRAS*	−0.35	−0.20	−0.19
*CXCL12*	−1.98	0.51	0.20
*CXCL14*	−2.17	−1.14	0.55
*CXCL2*	0.29	1.37	1.32
*CXCL3*	−0.98	0.81	0.31
*CCL2*	−2.52	−3.97	−1.27
*IL6*	−0.44	0.78	−1.28
*PDPN*	1.22	2.18	2.19
*CD29*	0.53	0.62	−0.11
*CD90*	0.47	−0.31	0.62
*Vimentin*	0.19	0.20	0.08
*GPR77*	0.01	1.05	−1.10
*CD10*	−0.12	0.69	0.36

## Data Availability

The datasets presented in this study can be found in online repositories. The names of the repository/repositories and accession number(s) can be found below: https://www.ncbi.nlm.nih.gov/geo/query/acc.cgi?acc=GSE277516 (accessed on 18 September 2024).

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
