# Peer review of "Extracellular Vesicles from Lung Adenocarcinoma Cells Induce Activation of Different Cancer-Associated Fibroblast Subtypes"

_biomedicines, 2024, doi:10.3390/biomedicines12112523_

Round 1
Reviewer 1 Report
Comments and Suggestions for Authors
This paper presented a sound rationale of how LUAD secreted EVs activated cancer-related fibroblast under different pathways and how they are differentiated from TGFβ pathway. The logic is sound, and evidence is sufficient. I suggest minor revision before publication of this paper
1. In some figures, the font size are different and hard to read. i.e. Figure 2 (a) to (c), font size are not consistent. Very same xy title and figure legend in 2(b)
2. The EV sample quantity was not clarified for the analysis of EVs secreted from 2073 and H1437 in Figure 2. Different cells secreted different amount of EVs. Do you control the same amount EV# or something else
3. One-way ANOVA should also be performed for some of the data in Fig.9 and 10
Author Response
Thank you very much for your thoughtful and constructive comments on our manuscript. We sincerely appreciate the time and effort you have taken to review our work. Your insights have been invaluable in helping us improve the quality of the paper, and we have carefully considered and addressed each of your suggestions, here is a point-by-point response to your comments:
- In some figures, the font size are different and hard to read. i.e. Figure 2 (a) to (c), font size are not consistent. Very same xy title and figure legend in 2(b)
Response: Thank you for the observation, we agree with this comment. The font size was adjusted in order to make the text more legible in Figures 1 (page 6) and 2 (page 7). Additional adjustment was made to Figure 2(b). We also increased the size of all the images which should also help.
- The EV sample quantity was not clarified for the analysis of EVs secreted from 2073 and H1437 in Figure 2. Different cells secreted different amount of EVs. Do you control the same amount EV# or something else.
Response: The number of EVs added per treatment were described in the results paragraph (on page 6, line 276) and the legend of Figure 2 on page 7, line 302. We added 7.13x1011 particles added for both LUAD cell lines. Indeed, different cells secreted different amounts of EVs but we did control the amount added for both treatments by quantitating the number of EVs added using Nanosight.
- One-way ANOVA should also be performed for some of the data in Fig.9 and 10.
Response: We performed a one-way ANOVA for both figures and these results were added to the text. Changes to figure 9 on page 15, figure legend line 1082, and change to the text on line 1052, page 13 and line 1395, page 19. Changes to figure 10 on page 16, figure legend line 1111, and change to the text on line 1127 of the same page.
Reviewer 2 Report
Comments and Suggestions for Authors
Comments to the Authors,
This is a review of the manuscript “Extracellular Vesicles from Lung Adenocarcinoma Cells Induce Activation of Different Cancer-Associated Fibroblast Subtypes” by Vazquez et al.
Dear authors, I believe your work but I have some concerns about it. Please see my comments below.
Abstract
The abstract is ok
Keywords
Keywords are ok
Introduction
I appreciate short and direct introduction sections, nonetheless, iu believe this one is slightly incomplete. The first paragraph, for example, is too direct. It bases the conclusion that new methods of diagnosis and treatment for lung cancer are necessary due to the number of cases per year, but what about the current state of the art? Current diagnosis procedures under use? Current treatments? These topics should be, at least briefly, addressed.
The theoretical part of introduction is ok but this chapter should have been concluded with a paragraph dedicated to summarizing the overall goal and tasks of the paper, as in wall scientific papers… something like “Considering all the aforementioned facts, this work aims at …”
Materials and Methods
No issues detected with this section
Results
The results are very interesting and can be useful to the academia, however, it is hard to assess their real impact on the overall work since each group of results is not properly discussed by the authors. The authors opted for separating the results form the discussing, hardening the understanding of the actual results. I would strongly recommend that the section dedication to materials/methods and discussion was merged into a single chapter. In this way, each group of results could be properly addressed and discussed by the authors enhancing the quality of the paper and the clarity of the results.
The bar graphs must contain the values of the error bars.
Table 1 has too many numbers, please consider the rules for significative numbers.
Discussion
Discussion is too long and hard to follow due to the necessity of constantly going back to consult the subchapter of results that is being addressed or discussed. I would recommend merging the sections.
A total of 49% plagiarism was detected. This must be fixed!
Author Response
Thank you very much for your thoughtful and constructive comments on our manuscript. We sincerely appreciate the time and effort you have taken to review our work. Your insights have been invaluable in helping us improve the quality of the paper, and we have carefully considered and addressed each of your suggestions, here is a point-by-point response to your comments:
- Introduction- I appreciate short and direct introduction sections, nonetheless, I believe this one is slightly incomplete. The first paragraph, for example, is too direct. It bases the conclusion that new methods of diagnosis and treatment for lung cancer are necessary due to the number of cases per year, but what about the current state of the art? Current diagnosis procedures under use? Current treatments? These topics should be, at least briefly, addressed.
The theoretical part of introduction is ok but this chapter should have been concluded with a paragraph dedicated to summarizing the overall goal and tasks of the paper, as in wall scientific papers… something like “Considering all the aforementioned facts, this work aims at …”
Response: Thank you for your insightful comments regarding the introduction section. We have added a brief overview of current diagnostic procedures and treatments for lung cancer to provide more context and highlight the need for new methods on page 1, in line 86 of paragraph 1 of the Introduction.
Additionally, we have concluded the introduction with a summarizing paragraph outlining the overall goals and tasks of the paper, in line with your recommendation on page 2, in line 34 of paragraph 6.
- Results- The results are very interesting and can be useful to the academia, however, it is hard to assess their real impact on the overall work since each group of results is not properly discussed by the authors. The authors opted for separating the results form the discussing, hardening the understanding of the actual results. I would strongly recommend that the section dedication to materials/methods and discussion was merged into a single chapter. In this way, each group of results could be properly addressed and discussed by the authors enhancing the quality of the paper and the clarity of the results.
Discussion- Discussion is too long and hard to follow due to the necessity of constantly going back to consult the subchapter of results that is being addressed or discussed. I would recommend merging the sections.
Response: After careful consideration, we have opted to retain the separation between these sections to maintain clarity and consistency with the journal's format. However, we fully acknowledge your concerns about the length and readability of the Discussion section. We have significantly condensed the Discussion to provide a more concise and focused interpretation of the results, ensuring that each finding is adequately addressed without overwhelming the reader. We removed the second paragraph of the discussion on page 17, in line 1160, the third paragraph on page 14, in line 1177, the last paragraph on page 18, in line 1356, the third paragraph on page 19, in line 1441and the fourth paragraph on page 19, in line 1442. We believe these revisions will improve the overall flow of the manuscript while addressing your valuable input.
- The bar graphs must contain the values of the error bars.
Response: For the PCR data in Figure 9 on page 15, we added in the same figure c) and d) tables per each graph with the RQ min and RQ max values. For the ELISA data in Figure 10 on page 16, we added in the same figure c) and d) tables per each graph with the error bar values representing the standard error of the mean (SEM).
- Table 1 has too many numbers, please consider the rules for significative numbers.
Response: Thank you for the observation, we agree with this comment. Extra decimals were removed in Table 1 page 8.
- A total of 49% plagiarism was detected. This must be fixed!
Response: A significant portion of the content of this article has high similarity with the first author’s master’s thesis, which was submitted to the University of British Columbia. The thesis has been cited in the article. Additionally, given the overlap, explicit permission from the respective university was asked before publication to ensure that all institutional requirements were met regarding intellectual property and the use of the thesis in further publications.
Round 2
Reviewer 2 Report
Comments and Suggestions for Authors
Dear Authors,
I believe you properly addressed my concerns. I propose acceptance. Congratulations.